# Shannon Entropy Loss in Mixed-Radix Conversions

**DOI:** 10.3390/e23080967

**Published:** 2021-07-27

**Authors:** Amy Vennos, Alan Michaels

**Affiliations:** Hume Center for National Security and Technology, Virginia Polytechnic Institute and State University, Blacksburg, VA 24061, USA; ajm@vt.edu

**Keywords:** PRNG, Shannon entropy, mixed-radix conversion

## Abstract

This paper models a translation for base-2 pseudorandom number generators (PRNGs) to mixed-radix uses such as card shuffling. In particular, we explore a shuffler algorithm that relies on a sequence of uniformly distributed random inputs from a mixed-radix domain to implement a Fisher–Yates shuffle that calls for inputs from a base-2 PRNG. Entropy is lost through this mixed-radix conversion, which is assumed to be surjective mapping from a relatively large domain of size $2^{J}$ to a set of arbitrary size *n*. Previous research evaluated the Shannon entropy loss of a similar mapping process, but this previous bound ignored the mixed-radix component of the original formulation, focusing only on a fixed *n* value. In this paper, we calculate a more precise formula that takes into account a variable target domain radix, *n*, and further derives a tighter bound on the Shannon entropy loss of the surjective map, while demonstrating monotonicity in a decrease in entropy loss based on increased size *J* of the source domain $2^{J}$. Lastly, this formulation is used to specify the optimal parameters to simulate a card-shuffling algorithm with different test PRNGs, validating a concrete use case with quantifiable deviations from maximal entropy, making it suitable to low-power implementation in a casino.

## 1. Introduction

The residue number system (RNS), initially proposed in 1959, was derived from the third century Chinese remainder theorem [1]. RNS architectures are now applied in many growing fields such as cryptography [2], image-processing systems [3], and error-correction codes [4] due to its convenience in parallel computing. Parallel processing with RNS often involves replacing a typical base-2 system with a different number representation system built upon two or more coprime number bases, which we refer to as mixed-radix (MR) [5].

Research in MR calculations in RNS implementations has increased due to applications of circuits in which radix choice affects their speed, power, and area [6]. In some implementations, such as those using low-power devices that require some level of security, binary-number representation may result in poor implementation or may not be applicable at all, requiring the consideration of another radix. For example, this distinction is apparent in Reed–Muller expansions over Galois fields involving cryptographic circuits [7]. In this application, using a lower-order radix usually requires less computation, which decreases the circuit’s area, but this technique allows for increased power consumption due to the large amount of interconnections. On the other hand, choosing a higher radix decreases power consumption, but increases the circuit area [8]. In the case that an optimal radix cannot be found, it is common to convert pseudorandomly generated words from one number base to another [9]. It is important to examine efficient ways to use MR techniques in processes such as Reed–Muller expansions. Researchers thus worked on independent MR conversion algorithms and techniques that have short conversion times [6,10].

MR techniques were also used for fast Fourier transform (FFT) pruning, which was designed to improve computational efficiency [11,12]. These algorithms have conventional applications, such as recording the flicker of voltage in smart homes [13]. Similar to the general circuit application, the choice of radix is important, and some radices may be better disposed in different scenarios depending on operation conditions. For example, higher radices reduce latency for memory-shared FFT architectures [14]. This additional use of MR conversions has prompted additional research on MR [12,15,16].

Most digital logic and computing systems are base-2, but algorithms such as Cooley–Tukey [17] offer equivalent mixed-radix representations to achieve the same overall calculation, but as a parallel composition of many small RNS operations. In particular, most PRNG methods were designed to produce bistate binary values, and methods to quantify PRNG output randomness usually require their binary form [18]. Consequently, the most common output of a PRNG is a *k*-bit binary word that may be viewed as an element $r\in Z_{2^{k}}$. Examples include linear feedback shift registers (LFSR) [19], the Mersenne Twister [20], and thermal entropy-based true random-number generators (TRNG) [21]. As such, the primary source domain for RNG values is typically on $Z_{2^{k}}$, while other algorithms consuming random words may wish for nonbinary random words. The specific example of optimally shuffling a deck of cards is discussed later on in this paper.

To support this evaluation, we focus on quantifying a specific MR application: calculating the Shannon entropy loss of an onto map from a source domain of size $2^{J}$ to a target domain of arbitrary size *n*. Ref. [22] developed a lower bound on the Shannon entropy loss from mapping between an RNS-based source domain onto a $Z_{2^{k}}$ target domain (a fixed *n* value), but this bound involves approximations that are only applicable for extraordinarily large source domains. In this paper, we calculate a more precise formulation that reveals the exact change in entropy of the onto map; our truncation of a Taylor series approximation offers a useable closed-form approximation that may be expanded by the reader. These calculations are referenced later to choose the correct parameter to reduce computations in our chosen application of card shuffling.

On the basis of these calculations, we apply MR conversions in a casino shuffling application with a base-2 PRNG component. As the casino industry is developing, casinos are relying on robotic card dealers to reduce the cost of hiring human card dealers and decrease the time that it takes to shuffle cards. Card-shuffling machines that randomize one deck while another is in use remove lost time and the possibility for fraudulent shuffles by a human dealer. These machines, which usually utilize a simple riffle shuffle, are rented for USD 500 per machine per month [23]. Additionally, a casino might benefit from the determinism of PRNG values to more precisely predict payouts; the same determinism becomes a potential avenue for dynamically skewing player odds [24], though the use of physics-based true RNGs can mitigate that risk.

Many prior works explored testing the quality of a card shuffle. More generally, card randomization is a popular problem in many fields such as statistics, combinatorics, and communications [25]. Markov proved results that analyzed card shuffles as early as 1906 using finite Markov chains [26]. More rigorous methods were also introduced, such as those using Fourier analysis and quantifying the entropy loss of riffle shuffling [27]. More recently, statistical tests for randomness, such as the FSU DIEHARD suite [28], were used to quantify the randomness of a permutation [29]. Additionally, entropy formulations, such as fiber entropy, were utilized to measure shuffle output randomness [30]. In our application, we determined the quality of the shuffle on the basis of Shannon entropy loss of the mapping process within the shuffle and simulating the shuffle algorithm using MATLAB with different PRNGs.

PRNGs and lightweight cryptographic primitives are useful in many applications. Though the option of using a TRNG in place of a lightweight primitive always provides a result with the highest entropy, utilizing PRNGs in these applications may be feasible and cost-efficient. In this paper, we demonstrate the feasibility of implementing a real-time, low-power card-shuffling algorithm with negligible entropy loss. The core algorithm is described in Section 2, followed by a complete derivation of Shannon entropy loss in Section 3. A simulation model is then presented in Section 4, followed by overall conclusions in Section 4.

## 2. Materials and Methods

To begin, we introduce a card-shuffling algorithm characterized by modulo arithmetic on RNG outputs and recurrent Fisher–Yates permutations [31]. The system, illustrated in Figure 1, shuffles a standard card deck as an ordered array C=[1|2|…|52] by repeated Fisher–Yates-based shuffles that utilize random numbers derived from any base-2 RNG.

Let *k* denote the width of the RNG output. Each clock cycle, *k* bits are produced by the RNG and are sent to the data-framing stage, which involves concatenating the bits from α PRNG outputs to create a word *X* of length J=α·k. The product of this stage is a word *X* that has enough bits to perform future modulo operations in the algorithm. For example, an 8-bit RNS with J=64 is framed collecting eight successive outputs and concatenating them together. Figure 2 demonstrates how *X* is framed in this step.

Once *X* is created, the *J*-bit string is mapped from a source domain of $Z_{2^{J}}$ to a dynamic target domain, *n* (starting at n=52), effectively producing a near-ideal uniformly distributed random value on $Z_{n}$. These modulo residuals are used in a Fisher–Yates permutation on C for each of the 51 iterations of the modulo calculation. Once the last Fisher–Yates shuffle is completed, deck C is fully shuffled.

One RNG may be better suited for this application than others are. in this case, “better” refers to both how closely the chosen RNG approaches maximal entropy, whether successive samples offer an opportunity to reverse-engineer, and the associated costs (money or computational complexity). As such, a TRNG would not contribute to distortions in output randomness, while utilizing a low-complexity PRNG may result in a noticeably nonrandom shuffle. However, the consumer of this algorithm may prefer the cost effectiveness of low-power PRNG instead of TRNG. This paper attempts to answer the underlying question of how to optimize the needs of both randomness and cost effectiveness.

### 2.1. Mixed-Radix Conversion Entropy

The shuffler’s deviation from maximal entropy occurs both through the PRNG algorithms that produce random input numbers and the onto mapping process of the mixed-radix conversion. In this section, we provide a background on Shannon entropy, the metric we used to measure entropy loss, and we explain how to calculate the Shannon entropy loss of a mixed-radix conversion from $Z_{2^{k}}\to Z_{n}$.

The Shannon definition of entropy quantifies the memoryless predictability in an event. In particular, the Shannon entropy of a discrete alphabet of possible events of size *L* is defined by
(1)$$H=-\sum_{i=1}^{L}p_{i}\log_{n}p_{i}.$$

We utilize Equation (Equation 1) to measure the entropy of modulo reductions A(modB), which can be represented by a surjective map A→B, where |A|=A and |B|=B. The mapping process can be visualized as placing the values 1,…,A into bins that represent their residual modulo *B*, creating a histogram of values as illustrated in Figure 3.

Splitting up the histogram on the basis of column height, we could calculate the Shannon entropy of onto map A→B. From this splitting, we tailor the Shannon entropy formula in Equation (Equation 1) to obtain
(2)$$\begin{array}{ccc} H & = & -\left(A\;\left(\mathrm{mod}\;B\right)\right)\left[\frac{1}{A}\left\lceil \frac{A}{B}\right\rceil \log_{B}\frac{1}{A}\left\lceil \frac{A}{B}\right\rceil \right]\\ & & -\left(B-A\;\left(\mathrm{mod}\;B\right)\right)\left[\frac{1}{A}\left\lfloor \frac{A}{B}\right\rfloor \log_{B}\frac{1}{A}\left\lfloor \frac{A}{B}\right\rfloor \right]. \end{array}$$

Further, define entropy loss for onto map A→B as shown in Equation (Equation 3).
(3)$$\gamma_{A\to B}=1-H\left(A,B\right).$$

The small entropy loss of a radix conversion indicates that more randomness is retained; likewise, this entropy-loss metric must always be positive and produce a value $0\leq \gamma_{A\to B}\leq 1$. We see the application of Equation (Equation 3) later in this paper, where entropy is calculated for mappings in which $A=2^{J}$ and *B* ranges from 2 to 52. Calculating these values is achieved by applying an adaptation of Equation (Equation 2).

### 2.2. Entropy Sources

Entropy is lost through the shuffler’s onto mapping process (discussed in the beginning of Section 2), and randomness is also lost through the chosen input RNG. Even though utilizing a TRNG in lieu of a PRNG or lightweight primitive provides the highest level of entropy, utilizing a PRNG in this application may be a feasible and cost-efficient choice. The concept of this experiment was, therefore, to build a card shuffler that is low-cost and low-power. We implemented the shuffling algorithm with different PRNGs and a TRNG to test their feasibility to create a random shuffle. These algorithms are listed and briefly described below.
*LFSR*: The linear feedback-shift register (LFSR) is a polynomial-based code on a binary space in which the output sequence is repeatedly replaced by an exclusive-or (XOR) of its last state. Since LFSR produces linear operations on a finite set of states, it creates a cyclic pattern. LFSR can produce a long cycle, or period, if the polynomial and input sequence is chosen correctly. Combined with its speed, this makes LFSR a reasonable choice for applications that require pseudorandom number generation. We used two LFSRs, LFSR-16 and LFSR-24, corresponding to polynomials $z^{16}+z^{15}+z^{13}+z^{4}+1$ and $z^{24}+z^{23}+z^{22}+z^{17}+1$, respectively.*Combined Multiple Recursive PRNG*: The combined multiple recursive PRNG combines multiple linear congruential generators, which are fast and low-cost PRNGs that use recurrence relations to obtain the next state. Linear congruential generator quality depends on choice of parameters, but their speed and low cost justified their use in cryptographic and statistical applications [32]. We used the MATLAB mrg32k3a variant, which has a period of $2^{191}$ [33].*Multiplicative Lagged Fibonacci*: a type of lagged Fibonacci generator (LFG) that was created in the 1950s in hopes of improving the linear congruential generator. These PRNGs utilize a recurrence formula on the basis of the Fibonacci sequence. A multiplicative LFG (MLFG) uses multiplication as the operation in question [34]. We used MATLAB’s mlfg6331_64, whose period is $2^{124}$ [33].*Mersenne Twister*: the default PRNG in simulation engines such as MATLAB and Python, the Mersenne Twister offers efficient speed and a large repetition period [20]. We used MATLAB’s mt19937ar, which has a period of $2^{19937}-1$, a Mersenne prime [35].Multiplicative congruential generator (MCG): Introduced in 1958, MCG is a specialized version of the linear congruential generator in which a parameter is set to zero [36]. Output quality passed randomness tests, but one must be careful in choosing some parameters [37]. We used mcg16807, which has a period of $2^{31}-2$ [33].Modified subtract with borrow generator (MSBG): MSBG was proposed in 1991 as an improvement to the lagged-Fibonacci PRNG [38], and later applied in the RANLUX generator created for particle-physics simulations [39]. We used MATLAB version swb2712, which has a period length of $2^{1492}$ [33].

After simulating the card shuffle using each of these RNGs, we analyzed the randomness of their outputs. We then discuss how RNG quality affects the resulting shuffle output. The goal is to uncover that, though a very low-power PRNG such as the LFSR may produce noticeable patterns within the output card shuffle, another lightweight primitive may be just as functional in producing a good shuffle as a high-cost TRNG. In order to explore these topics, we need to find an optimal testing value for the second parameter, *J*. This process is discussed in the following subsection.

### 2.3. Calculating an Optimal J

In this section, we find the value of *J* that should remain constant to test the feasibility and efficiency of the RNGs. The value of *J* should be large enough so that the algorithm’s output is shuffled to acceptable standards, but also small enough to minimize the amount of arithmetic performed in the modulo calculation. To determine a metric of shuffle quality, an engineer may write code and actually prove feasibility on a small device, while a mathematician could show some guarantee about the probabilities on the basis of the shuffles that we see. We employed a hybrid of these two perspectives, where we considered both the numerical Shannon entropy loss from the shuffle and the utilized hardware to implement the shuffle. The following subsections explore both of these approaches.

## 3. Results

### 3.1. Trend of $\gamma_{A\to \;B}$ When $A=2^{J}$

We visualized the entropy loss of the shuffler’s onto mapping process to notice any pattern in the behavior of $\gamma_{A\to B}$ with source and domain sizes that are relevant in this chapter. Our general frame of reference remains with an assumption that $2^{J}\gg n$, We created a simulation that tabulates the entropy loss for the onto map A→B, where $\left|A\right|=2^{J}$ and |B|=n as *n* ranges from 2 to 52. The result was a surface plot histogram, shown in Figure 4, of entropy loss values of J∈{6,7,…,64} and B∈{2,3,…52}. These entropy loss values were calculated on a deciBel scale to account for their size.

First, the *n* values that are a power of 2 (n∈{4,8,16,32,64}) have entropy-loss values of 0. This is because 2 raised to any power J≥6 modulo *n* equals 0; *n* automatically wraps around on an even basis, thereby creating zero residual. Visually, this plot appears to be monotonically decreasing with increasing *J*, yet the implications of that monotonicity represent a significant simplification in system design (i.e., *use the highest J value possible*) if true. Under the core assumption of $2^{J}\gg n$, we demonstrate this monotonicity, which is ultimately useful in our choice of *J*, since this statement justifies that choosing a larger value of *J* always results in lower entropy loss.

The derived approximation quantifies entropy loss, $\gamma_{2^{J}\to n}$ and $\gamma_{2^{J+1}\to n}$ of a surjective map A→B, in order to estimate the deviation from maximal entropy. The monotonicity conclusion falls apart if $2^{J}$ is not much greater than *n*, but it is utilized later in this chapter to determine a suitable value of *J* in a casino-game implementation depending on processor size. The simplifications of $H_{J}$ and $H_{J+1}$ are also helpful in that their algebraic form confirms a maximal entropy of 1 for $H_{J}$.

### 3.2. Approximation of Entropy and Entropy Loss for $Z_{2^{J}}\to Z_{n}$

**Claim**:Entropy loss as a function of *J* for our chosen ranges of (J,n) is a monotonically decreasing function when $2^{J}\gg n$. Additionally, the entropy, $H_{J}$, of a surjective map $Z_{2^{J}}\to Z_{n}$ under the condition of $2^{J}\gg n$ is(4)$$H_{J}\approx 1+\frac{2x^{3}-nx^{2}-n^{2}x}{2n{\left(2^{J}\right)}^{2}}+\frac{n^{3}x-3n^{2}x^{2}+4nx^{3}-2x^{4}}{2n{\left(2^{J}\right)}^{3}},$$where $x=2^{J}\;\left(\mathrm{mod}\;n\right)$. We also show $H_{J+1}-H_{J}\geq 0$ for all *n*, when $2^{J}>>n$.

**Proof**.Denote $H_{J}$ the entropy of the onto map A→B, where $A=2^{J}$ and B=n. Let $x=2^{J}\;\mathrm{mod}\;n$, where $2^{J}\gg n$. Using residual count *x*, the ceiling and floor values that occur in entropy value *H* are(5)$$\begin{array}{ccc} \left\lceil \frac{2^{J}}{n}\right\rceil & = & \frac{2^{J}}{n}+\frac{\left(n-x\right)}{n}, \end{array}$$(6)$$\begin{array}{ccc} \left\lfloor \frac{2^{J}}{n}\right\rfloor & = & \frac{2^{J}}{n}-\frac{x}{n}. \end{array}$$ For values including $2^{J+1}$ in the proof, we tabulate the following values in terms of *x*.(7)$$2^{J+1}\;\left(\mathrm{mod}\;n\right)=\left\{\begin{array}{cc} 2x, & \mathrm{if}x\leq \frac{n}{2}\\ 2x-n, & \mathrm{if}x>\frac{n}{2} \end{array}\right.$$(8)$$\left\lceil \frac{2^{J+1}}{n}\right\rceil =\left\{\begin{array}{cc} \frac{2^{J+1}}{n}+\frac{n-2x}{n}, & \mathrm{if}x\leq \frac{n}{2}\\ \frac{2^{J+1}}{n}+\frac{n-(2x-n)}{n}, & \mathrm{if}x>\frac{n}{2} \end{array}\right.$$(9)$$\left\lfloor \frac{2^{J+1}}{n}\right\rfloor =\left\{\begin{array}{cc} \frac{2^{J+1}}{n}-\frac{2x}{n}, & \mathrm{if}x\leq \frac{n}{2}\\ \frac{2^{J+1}}{n}+\frac{n-2x}{n}, & \mathrm{if}x>\frac{n}{2} \end{array}\right.$$ Then,(10)$$\begin{array}{ccc} H_{J} & = & -\frac{x}{2^{J}}\left(\frac{2^{J}+\left(n-x\right)}{n}\right)\log_{n}\left(\frac{2^{J}+\left(n-x\right)}{n2^{J}}\right)\\ & & -\frac{\left(n-x\right)}{2^{J}}\left(\frac{2^{J}-x}{n}\right)\log_{n}\left(\frac{2^{J}-x}{n2^{J}}\right). \end{array}$$ Using a degree-2 Taylor series approximation for the logarithms,(11)$$\begin{array}{ccc} H_{J} & = & -\frac{x}{2^{J}}\left(\frac{2^{J}+\left(n-x\right)}{n}\right)\left(\log_{n}\left(1+\frac{n-x}{2^{J}}\right)-1\right)\\ & & -\frac{n-x}{n}\left(\frac{2^{J}-x}{2^{J}}\right)\left(\log_{n}\left(1-\frac{x}{2^{J}}\right)-1\right) \end{array}$$(12)$$\begin{array}{ccc} & \begin{array}{c} = \end{array} & -\frac{x}{2^{J}}\left(1+\frac{2^{J}-x}{n}\right)\left(\frac{n-x}{2^{J}}-\frac{{\left(n-x\right)}^{2}}{2{\left(2^{J}\right)}^{2}}+\left(\sum_{i=3}^{\infty }{\left(-1\right)}^{i+1}\frac{{\left(n-x\right)}^{i}}{i{\left(2^{J}\right)}^{i}}\right)-1\right)\\ & & -\frac{n-x}{n}\left(\frac{2^{J}-x}{2^{J}}\right)\left(-\frac{x}{2^{J}}+\frac{x^{2}}{2{\left(2^{J}\right)}^{2}}-\left(\sum_{i=3}^{\infty }{\left(-1\right)}^{i+1}\frac{x^{i}}{i{\left(2^{J}\right)}^{i}}\right)-1\right). \end{array}$$The higher-order terms in the infinite summations of Equation (12) should be recognized as an alternating series that rapidly decays given our prior assumption of $2^{J}\gg n$. Via the alternating series remainder theorem [40], both residual summations are bounded by their third positive terms. With the extraction of these terms into a residual R(J,n) that incorporates the leading scalar coefficients, we may restate Equation (12) as(13)$$\begin{array}{ccc} H_{J} & = & -\frac{x}{n}\left(\frac{2^{J}+\left(n-x\right)}{2^{J}}\right)\left(\frac{n-x}{2^{J}}-\frac{{\left(n-x\right)}^{2}}{2{\left(2^{J}\right)}^{2}}-1\right)\\ & & -\frac{n-x}{n}\left(\frac{2^{J}-x}{2^{J}}\right)\left(-\frac{x}{2^{J}}+\frac{x^{2}}{2{\left(2^{J}\right)}^{2}}-1\right)\\ & & +R(J,n), \end{array}$$where(14)$$\begin{array}{ccc} \left|R(J,n)\right| & \leq & \left|-\frac{1}{3}\left(\frac{x}{n}\right)\left(\frac{2^{J}+\left(n-x\right)}{2^{J}}\right)\frac{{\left(n-x\right)}^{3}}{{\left(2^{J}\right)}^{3}}\right|\\ & & +\left|\frac{1}{3}\left(\frac{n-x}{n}\right)\left(\frac{2^{J}-x}{2^{J}}\right)\frac{x^{3}}{{\left(2^{J}\right)}^{3}}\right|\\ & = & o({\left(\frac{n}{2^{J}}\right)}^{3}). \end{array}$$In stating the little *o()* notation, we recognize that the scalar coefficients of each term are approximately $\frac{1}{3}$ each; thus, any additive combination between them is less than 1; likewise, x≤n, giving us an overall bound on residual term R(J,n). Further, given the magnitude of *n* in comparison to the magnitude of $2^{J}$ for our application, and the accompanying assumption that $2^{J}\gg n$, this residual term R(J,n) decays exceedingly quickly. If the assumption of $2^{J}\gg n$ is invalid, either additional terms must be retained in the Taylor series expansion and/or the chosen approach is invalid for drawing a conclusion. In the extreme case where $2^{J}≃n$, the onto map contains very few domain elements per row ($\left\lceil \frac{A}{B}\right\rceil$ described earlier); thus, the probabilities considered in the Shannon entropy estimation diverge from uniform distribution.This entropy value simplifies to(15)$$\begin{array}{ccc} H_{J} & = & 1+\frac{2x^{3}-nx^{2}-n^{2}x}{2n{\left(2^{J}\right)}^{2}}+\frac{n^{3}x-3n^{2}x^{2}+4nx^{3}-2x^{4}}{2n{\left(2^{J}\right)}^{3}}+o({\left(\frac{n}{2^{J}}\right)}^{3}) \end{array}$$Equation (Equation 15) is a key simplification of $H_{J}$. This form allows for us to simplify values in subtraction $H_{J}-H_{J+1}$. It remains to prove that $H_{J}$ is a decreasing function of J, for which we temporarily ignore the algebra of the bounded residual term. We broke that evaluation into two independent cases on the basis of the piecewise nature of Equations (Equation 7)–(Equation 9) for incremented case J+1.**Case 1 for $H_{J+1}$:**$x\leq \frac{n}{2}$. We used similar methods as those that we used in deriving Equation (Equation 15) to determine(16)$$\begin{array}{ccc} H_{J+1,x\leq \frac{n}{2}} & ∼ & 1+\frac{\frac{1}{2}n^{3}-\frac{3}{2}n^{2}x+nx^{2}}{2n{\left(2^{J}\right)}^{2}}+\frac{-\frac{3}{4}n^{4}+\frac{13}{4}n^{3}x-\frac{9}{2}n^{2}x^{2}+2nx^{3}}{2n{\left(2^{J}\right)}^{3}} \end{array}$$With further simplification and subtraction, we have that(17)$$\begin{array}{ccc} H_{J+1,x\leq \frac{n}{2}}-H_{J} & ∼ & \underset{\left(*\right)}{\underbrace{\frac{\frac{1}{2}n^{3}-\frac{3}{2}n^{2}x+nx^{2}-2x^{3}+nx^{2}+n^{2}x}{2n{\left(2^{J}\right)}^{2}}}}\\ & & +\underset{\left(**\right)}{\underbrace{\frac{-\frac{3}{4}n^{4}+\frac{13}{4}n^{3}x-\frac{9}{2}n^{2}x^{2}+2nx^{3}-n^{3}x+3n^{2}x^{2}-4nx^{3}+2x^{4}}{2n{\left(2^{J}\right)}^{3}}}}. \end{array}$$Given the prior assumption that $2^{J}\gg n$, (**) is arbitrarily small with practical bound on the order of $\frac{n^{3}}{{\left(2^{J}\right)}^{3}}$. This terms is also on par with the previously calculated R(J,n) residual. Consequently, we work with (*) to evaluate whether the difference of entropies is greater than 0. If we define $x=\frac{n}{2}+\epsilon$, letting ϵ be the residual between $\frac{n}{2}$ and *x*, $0\leq \epsilon \leq \frac{n}{2}$. Then,(18)$$\begin{array}{ccc} numerator\;of\;(*) & = & \frac{1}{2}\left(n^{3}-n^{2}x+4nx^{2}-4x^{3}\right)\\ & = & \frac{1}{2}\left(\frac{n^{3}}{2}-\epsilon^{2}n-2\epsilon^{3}\right). \end{array}$$So,(19)$$\left(*\right)=\frac{1}{4n{\left(2^{J}\right)}^{2}}\left(\frac{n^{3}}{2}-\epsilon^{2}n-2\epsilon^{3}\right).$$Since $\epsilon \leq \frac{n}{2}$, (*)≥0. Thus, $H_{J}$ is an increasing function as *J* increases. Consequently, entropy loss $1-H_{J}$ is a decreasing function in this case. The second case follows likewise, but we show the formulas for further clarification.**Case 2 for $H_{J+1}$:**$x>\frac{n}{2}$. The equivalent for Equation (Equation 16) is(20)$$H_{J+1,x>\frac{n}{2}}∼1+\frac{-\frac{1}{2}n^{2}x-nx^{2}+2x^{3}}{2n{\left(2^{J}\right)}^{2}}+\frac{\frac{1}{4}n^{3}x-\frac{3}{2}n^{2}x^{2}+4nx^{3}-2x^{4}}{2n{\left(2^{J}\right)}^{3}}$$We use simplification and subtraction similarly to Case 1 to reveal(21)$$\begin{array}{ccc} H_{J+1,x>\frac{n}{2}}-H_{J} & ≃ & \underset{\left(***\right)}{\underbrace{\frac{-\frac{1}{2}n^{2}x-nx^{2}+2x^{3}-2x^{3}+nx^{2}+n^{2}x}{2n{\left(2^{J}\right)}^{2}}}}\\ & & +\underset{\left(****\right)}{\underbrace{\frac{\frac{1}{4}n^{3}x-\frac{3}{2}n^{2}x^{2}+4nx^{3}-2x^{4}-n^{3}x+3n^{2}x^{2}-4nx^{3}+2x^{4}}{2n{\left(2^{J}\right)}^{3}}}} \end{array}$$As in the first case, (****) is substantially smaller than (***) on the order of $\frac{n^{3}}{{\left(2^{J}\right)}^{3}}$, which is comparable to the previously discussed R(J,n), so we worked with (***). We applied substitution $x=\frac{n}{2}-\epsilon$ and further simplifications as before. ϵ is still the residual between $\frac{n}{2}$ and *x*, $0\leq \epsilon \leq \frac{n}{2}$. We arrive at the analog of Equation (Equation 19),(22)$$\left(***\right)=\frac{1}{2n{\left(2^{J}\right)}^{2}}\left(\frac{n^{3}}{4}-\frac{n^{2}}{2}\epsilon \right).$$Since $\epsilon \leq \frac{n}{2}$, (***)≥0, and the discarded residual terms are sufficiently smaller than either difference in Equation (Equation 19) or (Equation 22), we conclude the monotonicity under the initial assumption of $2^{J}\gg n$. □

We end this subsection by emphasizing the importance of Equation (Equation 4). We simplified the entropy approximation into a closed form that approaches the value of 1. This value is relevant in that it represents the maximal entropy of $H_{J}$, and thus $0\leq H_{J}\leq 1$.

### 3.3. Applications and Relevance

Our reduction in Shannon entropy is both applicable for the card-shuffling application in this paper, and to measure entropy loss in other nonbinary applications. Binary modifications, such as gray code and the complex-binary-number system, are applied in puzzles [41], positioning technology [42], genetic algorithms [43], and granting faster processing for problems that handle complex numbers [44,45]. There are also many physical processes that are not necessarily optimized with radix 2. Research was conducted to examine the advantages of adopting the octal number system to represent SI units, money, time, and calendar days for computer accessibility [46]. Other systems include the decimal-number system, which is often utilized when high precision is necessary [47], and the alphanumeric number system, which is commonly utilized in storing colors using the RGB color model [48].

Some processes require a nonbinary source of randomness. PRNGs are designed to utilize nonbinary operations, such as the nonbinary Galois linear feedback shift register (LFSR), in which the exclusive-or performs addition modulo-*q* instead of modulo-2 [49]. In fact, some processes require a mix of different radices or mixed-radix as a source of randomness for PRNGs. This is common in instances in which there is interest in a uniform value on a specific domain size. In these cases, there is either the drawback of entropy loss that is not truly uniform from relying on a small PRNG, or the disadvantage of extra processing while utilizing a larger PRNG. One such example is mapping from one RNS space to a second coprime RNS space. Employing mixed-radix and nonbinary-number systems is useful in depicting metrics. For example, representing time in terms of years, days, hours, minutes, and seconds requires a number system that takes into account the cycle length of each unit. Currency is another example, in which we denote money in terms of dollars, quarters, nickels, and dimes [50], yet commensurate analysis would fail in fanciful analysis using prime-based currency (galleons, sickles, knuts). Generating a random number of seconds within a minute would not provide uniformity if it began with a binary PRNG; this may be performed with a conscious understanding of entropy loss given a *J*-sized processor.

Other applications of mixed-radix and nonbinary-number systems exist in games and combinatorial problems where the number base fluctuates throughout the processes. Though it may be simple to analyze the entropy of these games for one radix value, it is a much harder problem to understand entropy loss when the radix constantly changes, which is common in games. For example, researchers utilize the combinatorial number system to analyze lottery games, in which strictly decreasing combinations of numbers are advantageous to winning [51]. Another example occurs in video games, in which mixed-radix indexing processes are used to denote the position of different players [52]. Mixed radix is also necessary for combinatorial problems and simulations. For example, the factorial number system is a mixed-radix system that is utilized to analyze combinatorial problems in which permutations are represented as numbers [53]. It is important to convert formulae into input different radix values in these scenarios, so that others can understand the entropy loss of these games and problems. This paper highlights entropy loss in a card-shuffling system that can be applied to casino games: the index representing deck-position changes during every iteration of the system, so it is important to be able to calculate the entropy loss of the system with different radices.

Lastly, nonbinary and mixed-radix number systems are utilized in many scenarios that encompass even simple applications, such as representing metrics such as time and length. Utilizing nonbinary PRNG can help provide uniformity in generating random values for these applications. Utilizing RNS itself is also advantageous in parallel computing and fast arithmetic, which has applications in encryption, nanoelectronics, digital-image processing, and embedded computing. Our claim of Section 3.1 may be used to calculate the entropy loss of many of these mixed-radix and nonbinary applications, especially in games and combinatorial problems, because it is common for the number base to fluctuate throughout these processes. Equation (Equation 15), which quantifies the entropy loss of a surjective map A→B, where $\left|A\right|=2^{J}$ and |B|=n for an arbitrary value of *J*, is important in that it takes into account radix *n*. Thus, Shannon entropy loss can be calculated for multiple radix values in applications that utilize this mapping process.

### 3.4. Secondary Impact

In this section, we discuss parameters that influence our choice of *J* in testing PRNGs for the casino application. First, we must consider the impact of keeping *J* constant as *n* varies, or *mixing*
*J* values, such that we optimally select *J*, J(n), within an allowable range to minimize entropy loss for each *n*. By the approximation of Section 3.1, entropy loss is monotonically decreasing in terms of *J*. We want *J* to be as large as possible, and because of the monotonic nature of entropy loss, there is no apparent benefit in mixing *J* values. Therefore, we base our choice of *J* on the maximum that the processor can handle, regardless of current index value *n*.

Next, we consider hardware in choosing which *J* value to use in the simulations. In other words, we configure the choice of *J* to use available resources. Due to currently available standard processor sizes, the choices for *J* are restricted to J={8,16,32,64,128}. This is justifiable due to the monotonicity of entropy of $\gamma_{A\to B}$ when $A=2^{J}$. For example, if J=30, increasing this value to J=32 does not add any additional cost to the solution and would still result in lower entropy loss.

The fundamental hardware consideration in choosing the optimal *J* value depends on the processor size. The amount of arithmetic in the modulo process must not be too excessive for the *m*-bit choice of processor. The primary computation that would cause a large amount of arithmetic in the shuffle is computing a large *J*-bit value modulo *n*, where n≤52, on a *m*-bit processor when m<J. In the following paragraphs, we show that exceeding the processor constraints is not a realistic possibility for the shuffler, since many processors decompose large operations using bit-slice processes on a base-$2^{m}$ number system [54,55].

In the bit-slice process, the *J*-bit word *X* is divided into $\frac{J}{2^{m}}$ smaller *m*-bit subwords $X_{\frac{J}{2^{m}}-1},\ldots ,X_{1},X_{0}$. In our adaptation of a bit-slice processor [56], these subwords are used to compute X(modn) by Equation (Equation 23):(23)$$X=\left[\sum_{i=0}^{\frac{J}{2^{m}}-1}\left(X_{i}\;\left(\mathrm{mod}\;n\right)\cdot 2^{m\cdot i}\;\left(\mathrm{mod}\;n\right)\right)\right]\;\left(\mathrm{mod}\;n\right).$$

Each power of $2^{m\cdot i}$ modulo *n* can be precalculated and stored in memory to reduce the number of additional calculations. Figure 5 shows how an *m*-bit processor calculates Equation (Equation 23).

Since the *J*-bit word *X* is broken down into *m*-bit fractional elements, a small processor can compute each subword mod *n* to create a 6-bit or less output for each subword. Thus, even if J=128, which far exceeds entropy-loss expectations, a 32-bit processor suffices since it can decompose the word into four subwords. Important in this modular reduction is the recognition that subwords have virtually no pairwise calculations, making them efficient on virtually any reduced precision process.

Combining these secondary-impact types, we utilized values *J* equal to or larger than processor size *m* in our application for m≥32. If a smaller processor size was utilized, we used J=32 or higher. Table 1 displays the minimal value of *J* that is recommended for each common processor size.

### 3.5. Prototype Shuffling Algorithm

We used MATLAB to simulate the shuffling algorithm using the PRNGs listed in Section 2.2. The code that we utilized is outlined in Algorithm 1. Inputs to this algorithm include *k* and α as described in Section 2. The final input is array, which is an ordered set that can be mapped from a deck of 52 cards. This deck does not have to be in the order of a standard deck, yet we assumed a fresh deck between iterations in our experiment.

Block 1 creates a look-up table of precalculated values $2^{i}\;\mathrm{mod}\;n$ for i=1:α and n=1:52. These calculations utilize the bit-slice process described in Figure 5. Block 2 then calculates the random numbers by scaling PRNG value *r* with its corresponding value from *M*. Lastly, Block 3 implements the Fisher–Yates shuffle on array utilizing the random numbers from *X*. The output is a shuffled array that represents the final shuffled deck.

**Algorithm 1:** Shuffler Algorithm.

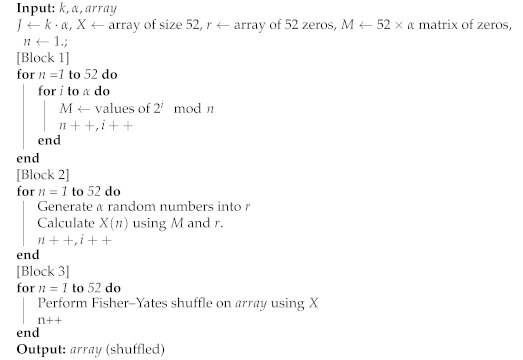



After simulating the shuffling algorithm with different PRNGs, we used a poker-hand ranker to evaluate the output shuffle quality [57]. Although not an explicit measure of security or goodness, this practical application of the shuffling algorithm enables testing the results beyond esoteric entropy numbers. The numbers of poker hands expected and obtained in a 10 million hand run, each consisting of an independent shuffle and dealing of the first 5 cards, are displayed in Figure 6. The calculated expected frequencies show that, even when the algorithm relies on moderate PRNGs, the output values are still randomized; weak PRNGs such as the LFSRs display an observable deviation from combinatorics-based expected probabilities. A TRNG may be used in lieu of any of these methods, providing greater assurance of maximal entropy.

## 4. Discussion

Due to the popularity of mechanical card shufflers in casinos, there is interest in creating real-time shuffling implementations utilizing lightweight primitives like PRNGs. We introduced a card-shuffling algorithm that, on the basis of RNG choice, can be low-power and cost-effective. We designed an experiment to test PRNGs of differing quality to determine how to optimize cost effectiveness and output shuffle quality.

Entropy is lost from this algorithm in two ways: through the onto-mapping process in modulo operations and through the selected PRNG. As an extension to the bound created by [22] for entropy loss resulting from the onto-mapping process, we calculated a more precise formula that describes the exact entropy loss. In particular, we quantified the Shannon entropy of surjective mappings A→B where |A|=A and |B|=B. After computing a generic formula for arbitrary domain sizes, we refined the formula for $A=2^{J}$. We utilized these formulas to prove the monotonicity of the onto map for $A=2^{J}$ and reasonable bounds on *n*. We created deterministic formulas and proven properties of the Shannon entropy of the onto map from $2^{J}\to n$. The motivation behind this proof was for choosing a suitable testing value of *J* in order to the functionality of different PRNGs in a casino shuffling algorithm.

We set up and proved the optimal parameters for testing to minimize entropy loss from the random-number-generation process in the algorithm. Then, we compared different PRNGs in the RNG process and examined the effect of using a low-power PRNG on the entropy of the combined operation by examining the frequencies of poker hands out of 10 million runs for each PRNG, listed in Figure 6. The resulting frequencies demonstrated that even low-power PRNGs are able to produce a suitable amount of randomness for a casino shuffling application, though we also showed that not all PRNGs make the cut.

Future work is building a hardware prototype of this shuffling engine, including the incorporation of a TRNG. Applying poker-hand analysis with TRNG is expected to provide long-term probability values for comparison that can truly highlight the differences in utilizing PRNG in lieu of TRNG for this application. Moreover, the prototype is suitable for immediate incorporation in electronic games (e.g., video poker and slots), which may require different number bases and adaptation to mechanical shufflers.

## Figures and Tables

**Figure 1 entropy-23-00967-f001:**
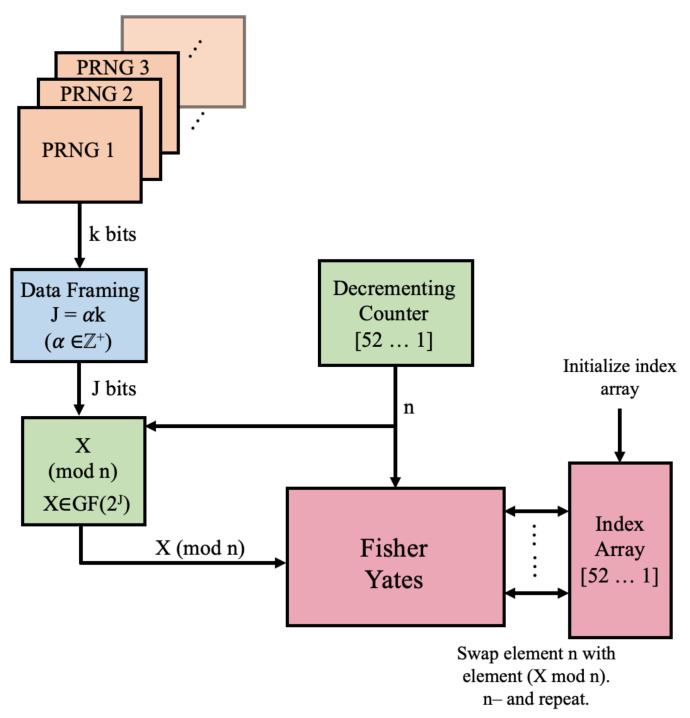
Shuffler diagram.

**Figure 2 entropy-23-00967-f002:**
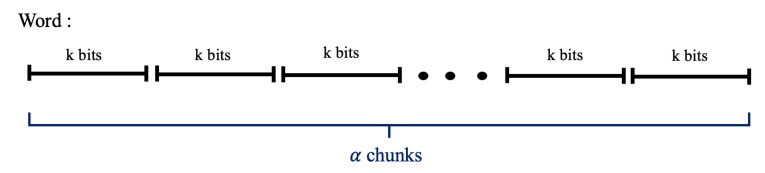
Data-framing stage concatenates α words, each composed of *k* bits, to create a *J*-bit string.

**Figure 3 entropy-23-00967-f003:**
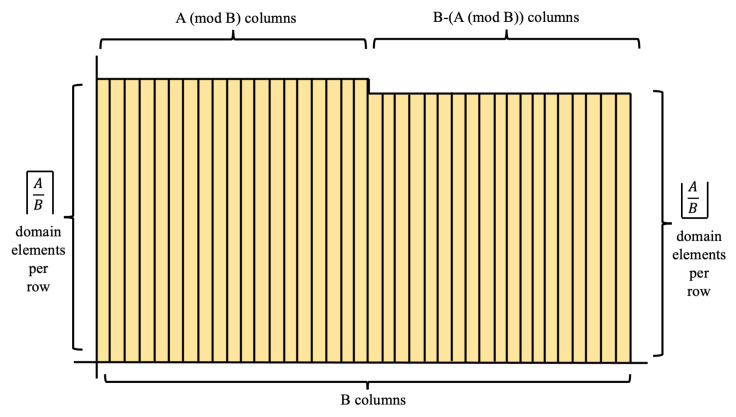
Histogram of residuals resulting from surjective mapping A→B, where |A|=A and |B|=B.

**Figure 4 entropy-23-00967-f004:**
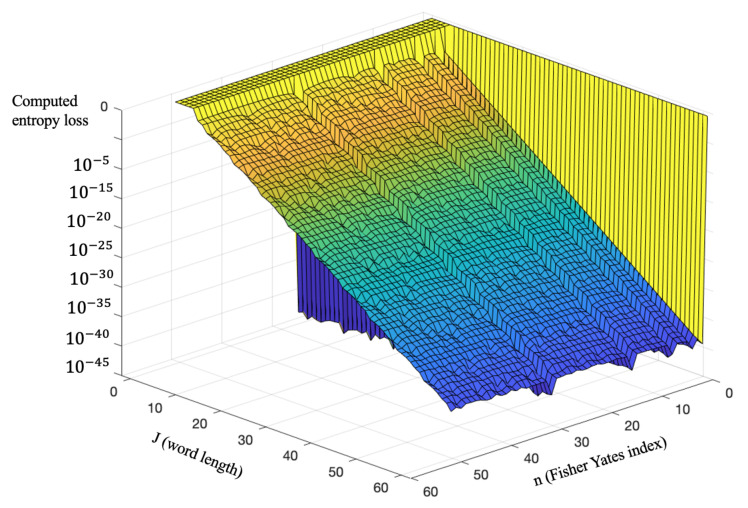
Entropy-loss values as *J* in the range of 6–64, and as B in the range of 2–52.

**Figure 5 entropy-23-00967-f005:**
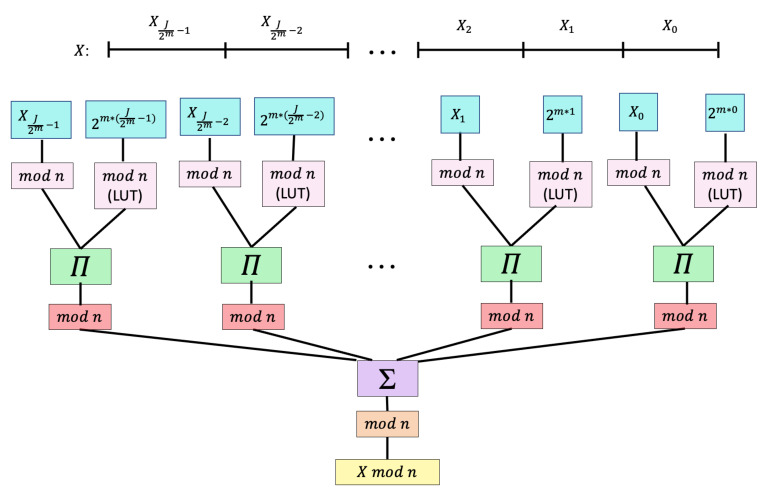
Bit-slice process of decomposing the calculation of *J*-bit binary word modulo *n* on a *m*-bit processor.

**Figure 6 entropy-23-00967-f006:**
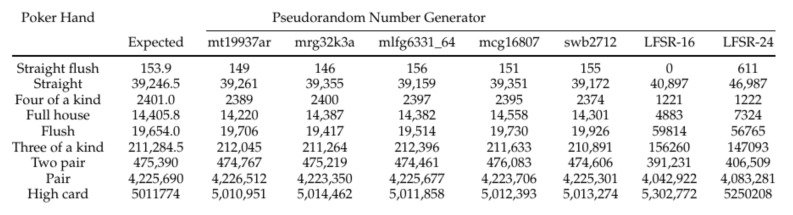
Number of events of 9 poker hands out of 10 million runs for each tested PRNG.

**Table 1 entropy-23-00967-t001:** Choice of *J* for card-shuffling implementations based on processor size.

Processor Size *m*	Minimal Choice of *J*
8	32
16	64
32	128
64	128
128	128

## Data Availability

This study did not report any data.

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
