# Peer review of "Shannon Entropy Loss in Mixed-Radix Conversions"

_entropy, 2021, doi:10.3390/e23080967_

Round 1
Reviewer 1 Report
The paper focuses on entropy loss during Mixed radius conervsions with a particular focus on 52-card decks shuffling (using FIsher-Yates Algorithms).
Section 1 is an introduction including a quite complete related work part. The problematic is well exposed and the presentation is motivating.
Section 2.1 prodvides the theoretical notions and a formal description of the problem.
Section 2.2 describes the possible sources of entropy, mainly based on classical PRNG.
Main theoretical results are provided in Section 3. As I will explain bellow, this section requires some clarification.
Section 3.1.1 focuses on hardware impact. It is out of my skill scope.
Experimental results as well as the algorithm proposed in Section 3.2 are interesting. Some signtificant differences/bias in PRNG are pointed out.
-----------------------
Major Comments
The paper is well written and motivated. I have to connected main issues with the paper.
1) The first one (that can be easily solved) it that the equality (14) is not true but is up to o(..). A t Taylor expansion is not an equality and (11) is not equal to (12). Theorem 1 has to be rewriten into an exact one using small o().
2) Considering the previous point, all the equations (14), (15),... are not exact. I don't know how it's impact the proof. Looking at Figure 4, I'am not sure that Hj is mathematically a decreasing function of J (even if globally it's decreasing). Maybe it's a 3D effect and Hj is deacring on Fig. 4. But I think it has to be proved rigourously.
------------------------
Minor comments/suggestions
- Page 4, line 140. Change leq into \leq.
- Page 5, line 146. There is a missing section number (either bad \ref{} or a missing latex compilation)
- Page 7, line 13. It's better to avoid mathematical quantificators in text sentences. I suggest to replace the \forall into "for all"
- Page 7, equations (7), (8), (9). Add "if" before "x<= n/2", "x > n/2"
- Page 7, footnote: write it in a mathemacial sound way to use it explicitly in the proof when it is required.
- Page 8, line 231. I suggest to add a sentence of the kind "It remains to prove that Hj is a decreasing function of J."
- Page 9, page 247. It's strange to have s Subsection 3.1.1 without a Subsection 3.1.2.
- Page 13, line 420. Isn't there a capital letter missing at the beginning of "chih" (I don't know, I am not an expert in Chinese names)
Author Response
We would like to thank you for your useful comments and suggestions. Please see attachment.

Reviewer 2 Report
Paper title: Shannon Entropy Loss in Mixed-Radix Conversions
Reference No.: entropy-1243459, to the journal “Entropy ”
This paper can be accepted after considering the following points:
- The reference of Eq.1 must be stated.
- P.5 L. 146, “….discussed in Section ??). what is meant by “??”.
- The reference of Eq.22 must be stated.
- The authors need to compare the numerical results with other publishing one to show the accuracy of the calculations or compare with some special cases.
- The authors need to state some applications (especially physical applications) of their results.
Author Response

(The authors gave the same response as above.)

Reviewer 3 Report
The results presented in the paper are very interesting. I have no objections to the results shown. I suggest that the paper be accepted in its existing form.
Author Response
We would like to thank you for your report.
Round 2
Reviewer 1 Report
The report is in the enclosing file.
Best regards

Author Response
We would like to thank you for your suggestions. We have addressed them in the attached letter and have highlighted updates in the newly submitted paper.

Round 3
Reviewer 1 Report
Dear authors,
This third version of the paper is now worth for publication in my opinion.
I have two minor suggestions
- In (15), the \sim can be replaced by = since there is the o().
- For the last sentance of the claim, I think it would be better to write explicitely "We also show H_{j+1}-H_j \geq 0 for all n, when 2^J >>n".
Best regards